# CAPG Is Required for Ebola Virus Infection by Controlling Virus Egress from Infected Cells

**DOI:** 10.3390/v14091903

**Published:** 2022-08-28

**Authors:** Hiroyuki Mori, James P. Connell, Callie J. Donahue, RuthMabel Boytz, Yen Thi Kim Nguyen, Daisy W. Leung, Douglas J. LaCount, Robert A. Davey

**Affiliations:** 1Department of Microbiology, NEIDL, Boston University School of Medicine, Boston, MA 02215, USA; 2Department of Medicinal Chemistry and Molecular Pharmacology, Purdue University, West Lafayette, IN 47907, USA; 3Department of Pathology and Immunology, Washington University School of Medicine, St. Louis, MO 98195, USA; 4Department of Medicine, Washington University School of Medicine, St. Louis, MO 98195, USA

**Keywords:** Ebola virus, actin, egress, assembly, trafficking, host interaction

## Abstract

The replication of Ebola virus (EBOV) is dependent upon actin functionality, especially at cell entry through macropinocytosis and at release of virus from cells. Previously, major actin-regulatory factors involved in actin nucleation, such as Rac1 and Arp2/3, were shown important in both steps. However, downstream of nucleation, many other cell factors are needed to control actin dynamics. How these regulate EBOV infection remains largely unclear. Here, we identified the actin-regulating protein, CAPG, as important for EBOV replication. Notably, knockdown of CAPG specifically inhibited viral infectivity and yield of infectious particles. Cell-based mechanistic analysis revealed a requirement of CAPG for virus production from infected cells. Proximity ligation and split-green fluorescent protein reconstitution assays revealed strong association of CAPG with VP40 that was mediated through the S1 domain of CAPG. Overall, CAPG is a novel host factor regulating EBOV infection through connecting actin filament stabilization to viral egress from cells.

## 1. Introduction

Ebola virus (EBOV) is a cause of severe hemorrhagic disease with a high mortality rate in humans and non-human primates [1]. Sporadic outbreaks, including the widespread 2013–2016 outbreak, have threatened human health and society significantly, with EBOV continuing to be a global health burden [2]. Even though much molecular and clinical research has been conducted to develop promising therapeutics, only a limited number of treatments and vaccines have been approved for clinical use [2]. These therapeutics must be administered early in disease onset to be effective and are difficult to deploy in outbreak conditions. The limited availability of countermeasures as well as the risk of viral resistance emphasizes the need for further therapeutic development. Since the virus replication cycle depends on the host cell and cellular machinery, host factors provide attractive targets for alternative antiviral drug development. Thus, understanding the molecular mechanisms of EBOV infection in host cells and the subsequent interactions of host and virus factors is necessary for combating future outbreaks and disease.

EBOV is a non-segmented, negative-sense RNA virus belonging to the family *Filoviridae*. Virions are morphologically characterized by diameters of ~80 nm and lengths ranging from hundreds of nanometers to micrometers [3]. The filovirus genome encodes nine proteins, with seven present in intact virions and two truncated glycoproteins that are secreted [4]. The viral nucleocapsid is composed of the RNA genome in association with the nucleoprotein (NP), the viral polymerase L, and replication and transcription factors VP35 and VP30 [5]. VP40, the viral matrix protein, contains a PPXY motif capable of interacting with a variety of host cell proteins [6] and is the most abundant protein in the virion [4]. During late stages of virus assembly, VP40 connects the viral nucleocapsid to the host-membrane derived viral envelope [6], in which the viral glycoprotein GP is embedded [3]. GP serves as the major mediator of viral entry into cells [7].

EBOV enters cells via macropinocytosis [8,9], a cellular process heavily dependent on actin function [10]. Macropinocytosis is driven by membrane ruffling, which is mediated by cortical actin branching at the cytosolic surface of the plasma membrane [10]. This actin remodeling is dependent on the actions of Rho-GTPases such as Rac1 and Cdc42, in addition to the multimeric Arp2/3 complex, all of which are regulated upstream by PI3 kinase (PI3K) signaling pathway [10]. We and others previously showed that inhibition of PI3K and Rac1 interferes with cellular uptake and entry of EBOV [11] and additional studies implicated the involvement of additional Rho–GTPases in EBOV entry [12], ultimately demonstrating a heavy involvement of actin dynamics in successful viral entry and overall infection.

Actin polymerization has also been shown to participate in late stages of EBOV infection, notably in assembly and egress [13]. Assembly initiates with the formation of the nucleocapsid, a process driven by NP, VP35, and VP24 [14]. VP24 facilitates viral transcription and replication and ensures packaging of the viral genome into the nucleocapsid [15,16]. The nucleocapsid is then transited to sites of VP40-driven viral budding at the plasma membrane via actin polymerization [13], a process that is again dependent on actin nucleation by Arp2/3 [17]. VP40 is capable of inducing membrane curvature and can independently induce viral budding [18], though budding is enhanced with expression of GP. Through interaction with host factors TSG101 and VPS4, VP40 coordinates release of viral particles [19]. Like nucleocapsid transit, VP40-driven budding is also dependent on actin-functionality [20], with actin being detected within VP40 budding protrusions and virus-like particles (VLPs) [21]. Inhibitors of calmodulin, a calcium sensing protein capable of regulating filamentous actin (F-actin) formation in cells [22], similarly inhibits VP40 driven budding and egress [13]. Successful production of infectious virions therefore is closely connected with actin processing and production.

Actin remodeling is controlled by interactions with actin-interacting proteins that regulate actin monomer polymerization to F-actin, depolymerization and branching. One such protein is macrophage-capping protein (CAPG), a member of the Gelsolin/Villin family that binds to the barbed-end of F-actin and responds to Ca^2+^ and phosphoinositide signals (Appendix A). The main function of this family of proteins is to promote stability of the actin filament by capping the growth end of the main filament and branches to prevent further elongation while preventing loss of actin subunits [23]. CAPG is unique among its family members as it lacks actin severing activity and consists of only three gelsolin homology (GH) domains while others have six [24]. The role of CAPG and the Gelsolin/Villin family in macropinocytosis, as used by EBOV to enter cells, is not fully understood. However, it has been shown that CAPG and gelsolin double-null mice exhibit diminished cell membrane ruffling and protrusion, which are characteristics of macropinocytosis [25].

Here, we studied the role of CAPG on EBOV infection for cell entry, replication, and budding from infected cells. We show that CAPG plays only a minor role in virus uptake into cells. Instead, it plays a predominant role in virus egress from cells. This appears to be mediated by close association with the major nucleocapsid protein, VP40.

## 2. Materials and Methods

### 2.1. Cells and Culture

Human cervical carcinoma (HeLa) cells and Vero E6 cells (ATCC, Manassas, VA, USA) were cultivated in culture medium: Dulbecco’s modified Eagle’s medium (DMEM) (11995073; Gibco, Gaithersburg, MD, USA) supplemented with 10% fetal bovine serum (FBS) (S10350; R&D systems, Minneapolis, MN, USA) and 1% penicillin-streptomycin solution (15140-122; Gibco). All cells were maintained at 37 °C in a humidified incubator with 5% CO_2_.

### 2.2. Reagents and Antibodies

Hoechst 33342 dye (#62249) and HCS CellMask blue (#H32720) for immunostaining were purchased from ThermoFisher Scientific (Waltham, MA, USA) and used at 1:10,000 and 1:1000 in PBS, respectively. Mouse monoclonal anti-ZEBOV GP (4F3) and anti-ZEBOV VP40 (3G5) antibodies were obtained from IBT Bioservices (Gaithersburg, MD, USA). Mouse monoclonal anti-ZEBOV NP (10B5 C11 C9 C3), and VP35 (2A11 F12 D7) were obtained by Dr. D. Leung (Wash. U. St. Louis, MI, USA). Rabbit polyclonal anti-Ebola VP30 (GTX134035) was purchased from GeneTex (Irvine, CA, USA). Rabbit polyclonal anti-CAPG antibody (10194-1-AP) was from Proteintech (Rosemont, IL, USA). Mouse monoclonal anti-*β*-actin (MAB8929) antibody was from R&D systems. The following secondary antibodies were purchased from Thermo Scientific: goat anti-mouse Alexa Flour 488, anti-mouse Alexa Flour 546. 5-(N-ethyl-N- isopropyl) amiloride (EIPA) and chlorpromazine hydrochloride (CPZ) were from Sigma-Aldrich (St. Louis, MO, USA). EIPA was diluted to 10 mM with dimethylsulfoxide. CPZ was diluted to 50 mg/mL with water. All the reagents described here were kept at −40 °C until use.

### 2.3. Plasmids

Plasmids were cultivated in E. coli DH5α, purified by PureLinkTM HiPure Plamid Midiprep kit (K210004; ThermoFisher Scientific), Plasmid Maxi kit (12162; Qiagen, Germantown, MD, USA) or ZymoPURE II Plasmid Midiprep Kit (D4200; Zymo Research, Irvine, CA, USA). pCAGGS-VP40, pCAGGS-enhanced green fluorescent protein (eGFP)-VP40, pcDNA3-GP, pCAGGS-NP, pCAGGS-VP30, pCAGGS-VP35, and pCAGGS-L constructs are described elsewhere [26,27,28]. Each plasmid expresses the Mayinga variant of the indicated EBOV protein. For the tripartite split-GFP system, GFP B10 and B11 fusion constructs were created in pCDNA3- and pCDNA5-based plasmids [29]. GFP B10 was fused to EBOV VP40 and VP30 at the 5’ and 3’ ends, respectively, whereas GFP B11 was fused to human genes or gene fragments (5’ fusions for CAPG, TSG101, and utrophin CH domain; 3’ fusions for CAPG and PABPC1). Genes were PCR-amplified with primers listed in Appendix A and inserted by ligation or NEBuilder HiFi DNA Assembly (E2621; NEB, Ipswich, MA, USA), or transferred from another plasmid by restriction digest and ligation. All plasmids were confirmed by DNA sequencing. Plasmid maps and sequences can be found in Appendix A. Primer sequences and cloning details are available upon request. pCDNA3-GFP1-9 T2A mCherry was a gift from Xiaokun Shu (plasmid # 124430, Addgene, Watertown, MA, USA) [30]. Additional details can be found in the Appendix A and Methods.

### 2.4. Virus Infections

All experiments with replication-competent EBOV were performed in a biosafety level 4 (BSL4) laboratory at National Emerging Infectious Diseases Laboratories, Boston University (Boston, MA, USA). All infection assays were performed in culture medium and in at least triplicate. HeLa cells were seeded in 96, 24, 12 or 6 well plates (3596, 3524, 3512, 3506, respectively; Corning, Glendale, AZ, USA) and incubated overnight at 37 °C. Only wells showing 50–70% confluency of cells were used for subsequent infection assays. The cells were then challenged with wild type (WT)-EBOV or EBOV-GFP and incubated for 1 h at 37 °C. The supernatant was removed and cells were washed twice with DMEM without serum. Fresh DMEM supplemented with 10% FBS was added and cells were incubated for the appropriate time depending on the assay to be performed. The MOI for all infections was 0.2 unless otherwise indicated.

### 2.5. Calculation of Infection Efficiency

At the indicated times after infection, cells were fixed with 10% neutral buffered formalin (NBF; LC146705; ThermoFisher Scientific) overnight at 4 °C. For experiments using GFP-EBOV, cells were stained with Hoechst 33342 dye for 30 min at room temperature. Fixed cells were permeabilized in 0.2% Triton X-100 detergent (215682500; ThermoFisher Scientific), and blocked with 3.5% Bovine Serum Albumin (BSA) (BP1600; ThermoFisher Scientific) for 1 h at room temperature. After washing twice with PBS, the cells were stained with anti-EBOV GP antibody (4F3) diluted 1:1,500 in 3.5% BSA for 1 h. A goat anti-mouse secondary antibody conjugated to Alexa Fluor 488 was diluted 1:2000 in 3.5% BSA and incubated with cells for an additional 1 h. Nuclei were stained with Hoechst 33342 dye used at 1:10,000. Plates were imaged using a Cytation 1 multimode plate reader (Biotek, Winooski, VT, USA). Multiple non-overlapping images were taken for each well to ensure at least 5000 cells were captured. The number of GFP or GP-positive cells and nuclei in each well were counted using CellProfiler (Ver. 4.2.1; Broad Institute, Cambridge, MA, USA) pipelines developed by the authors (available on Zenodo [31]). The infection efficiency, defined as the percentage of GFP- or GP-positive cells per well, was determined by dividing the number of positive cells by the total cell nuclei count. Reported means and standard deviations are for at least three biological replicates.

The yield assay for measuring viral particle release from cells was performed using Vero E6 cells incubated with 8-fold serial dilutions of the cell culture supernatant from samples at 37 °C for 48 h. Virus release efficiency, defined as the number of GP-positive cells per total cell nuclei, was calculated as described above. Results are reported as the infection efficiency per mL of inoculated supernatant.

### 2.6. Small Interfering RNA (siRNA)

siRNAs targeting exons of CAPG were purchased from Qiagen (Appendix A). AllStars Negative Control siRNA (non-silencing siRNA; 1027281) was purchased from Qiagen and used as a negative control. HeLa cells were transfected with each respective siRNA at indicated concentrations using RNAiMAX (13778075; ThermoFisher Scientific) following the manufacturer’s protocol. After a 48 h incubation, the cells were either challenged with virus in the BSL4 laboratory or transfected with plasmids encoding cell or virus proteins.

### 2.7. Generation of Knockout (KO) Clonal Cell Lines

Lentiviral vectors encoding CAPG sgRNAs (clone ID HS5000023627 GCTCATCCCGGGATGACTGCTGG; clone ID HS5000023628 GTTGAGGTGCACAGCCAGCACGG, Sigma-Aldrich) were transduced into HeLa cells at MOI of 1.0 with 8 µg/mL of polybrene (Sigma-Aldrich) by centrifuging cells and vectors at 2300 × rpm for 1 h at 22 °C. After further incubation with 5 µg/mL puromycin (A11138-03; Gibco) for 2 h at 37 °C, cells were washed with PBS, then incubated in fresh DMEM+10% FBS. 48 h post transduction, cells were trypsinized and seeded into 96 well plates (1.0 × 10^4^ cells/well) in DMEM+10% FBS supplemented with 20 µg/mL G418 to select for transduced cells. Selection was repeated for 6 passages and selected cells were expanded into 24 well plates and transfected with pX330-U6-Chimeric_BB-CBh-hSpCas9 (#42230; Addgene, Watertown, MA) by TransIT-LT1 (Mirus Bio, Madison, WI, USA). Single cell clones were established by limiting dilution. Knockout was confirmed by immunoblotting and genetic changes identified by Sanger sequencing (performed by GENEWIZ, NJ, USA).

### 2.8. Cell Viability Assay

The effect of CAPG depletion on viability of KO and siRNA-treated cells was assessed at several time-points using an MTT assay (11465007001; Sigma-Aldrich) per manufacturer directions. Briefly, 10 µL of MTT labeling reagent was added to each well and incubated for 4 h at 37 °C. 100 µL of solubilization buffer was added and incubated for additional 1 h. Absorbance at 575 nm with a reference wavelength of 675 nm was measured using a Tecan Spark plate reader V2.1 (Tecan, Switzerland). Cell viability for each well was normalized by subtracting the reference measurement from the 575 nm absorbance value.

### 2.9. Virus-Like Particle Release Assay

HeLa cells were seeded in 6 well plates (2.0 × 10^5^ per well) and incubated overnight before transfection with pCAGGS-VP40 using TransIT-LT1 (MIR2304; Mirus Bio) according to the manufacturer’s protocol. Cell lysates and supernatants were harvested 48 h after transfection. Supernatant samples (2 mL each) from each well were harvested and clarified by low-speed centrifugation (3000× *g*, 15 min). The clarified supernatant was further concentrated and purified over a 20% sucrose cushion in a 60 mm polypropylene tube (328874; Beckman Coulter, Brea, CA, USA), which was spun in an ultracentrifuge at 30,000× *g* for 2 h at 4 °C (SW 60 Ti rotor, Beckman). The pellet was resuspended in 20 µL of PBS for 1 h at 4 °C, and viral protein content was assessed by immunoblot. The remaining cells were lysed using RIPA buffer (BP-115; Boston BioProducts, Milford, MA, USA) and used in immunoblot assays.

### 2.10. Immunoblot Assay

Cell lysates were prepared as follows: After removing medium and washing twice with PBS, an appropriate volume of RIPA buffer with protease inhibitor (A32965; ThermoFisher Scientific) was added to adherent cells. The cells in the buffer were disrupted by repeated freeze–thaw cycles and pipetting. The lysates were cleared by centrifugation and supernatants were collected for subsequent immunoblotting. Lysate samples from EBOV-infected cells were removed from the BSL4 following virus inactivation. Briefly, SDS was added to the cell lysate to a final concentration of 1%, and the mixture was boiled for 10 min. Immunoblotting used a Jess capillary electrophoresis system (ProteinSimple, San Jose, CA, USA) according to the manufacturer’s recommendations. β-actin was used for loading standardization between lanes. Band intensities were measured using the Compass software (Protein-simple, Biotechne) and analyzed using Fiji software [32] Ver. 2.3.0.

### 2.11. Immunofluorescence Based Detection of Virus Infection and Uptake of Virus into Cells

For immunofluorescent assays, cells were fixed with 10% neutral buffered formalin overnight at 4 °C, then permeabilized with 0.2% TritonX-100 for 15 min. After blocking with 3.5% BSA, cells were stained with respective primary antibodies for 1 h at room temperature. Following PBS washes, secondary antibodies were added for an additional 1 h. The cells were further stained with HCS CellMask Blue to visualize both cell bodies and nuclei. After washing with PBS, the cells were imaged using a Ti2 Eclipse microscope (Nikon, Tokyo, Japan) with a 100× oil immersion lens.

Virion internalization assays were performed as previously described [26,33]. Briefly, HeLa cells were seeded in 8-well chamber slides (80826; ibidi, Gräfelfing, German) at 2.0 × 10^4^ cells per well and incubated with EBOV for 1 h at 14 °C to synchronize binding of viral particles to the cell surface without uptake. Immediately after the incubation, cells were washed three times with cold PBS, then incubated in fresh DMEM with 10% FBS for 6 h at 37 °C. After fixation with 10% formalin, the cells were blocked in 3.5% BSA and stained with the anti-GP antibody (4F3) for 1 h without permeabilization. A goat-anti mouse secondary antibody conjugated to Alexa Fluor 594 was used to label membrane-bound GP. Then, cells were permeabilized, blocked again, and restained with the same anti-GP antibody, followed by incubation with Alexa Fluor 488-conjugated goat anti-mouse IgG antibody. The cell body and nuclei were stained with HCS CellMask blue. Images were taken at multiple z-planes on a widefield Nikon Ti2 and deconvolved to resolve objects in 3D using Microvolution software (Microvolution Inc., Cupertino, CA, USA) run on Fiji [32]. The deconvolved images were analyzed using Imaris 3D image analysis software (Bitplane Inc., Belfast, UK). Green foci and overlapping green-red foci were counted. Virus internalization efficiency was calculated as the ratio of green foci to total foci.

### 2.12. Proximity Ligation Assay

The Duolink Proximity Ligation Assay kit (Sigma-Aldrich) was used to observe proximity between viral and host proteins inside cells. In brief, HeLa cells were seeded onto 18 well chamber slides (81816; ibidi) at 0.5 × 10^4^ cells per well. After incubation overnight, the cells were infected with WT-EBOV for 24 h. The cells were fixed and permeabilized, and the proximity ligation was performed following the manufacturer’s protocol using specific primary antibodies described above. Actin filaments were stained with Alexa Flour 488-conjugated phalloidin (A12379; ThermoFisher Scientific) for 30 min. Red foci inside cells indicated that two targeted proteins were in close proximity. Images were taken at 40× or 100× oil immersion lens on a widefield Nikon Ti2 microscope. Z-stacks were deconvolved and used to resolve objects in 3D, as above, and used to detect protein interactions with phalloidin-labeled actin filaments.

### 2.13. Tripartite Split-GFP Assay

HeLa cells were transfected with GFP B10, GFP B11 and GFP1-9-T2A-mCherry expression plasmids at a ratio of 1:5:5 (typically 15 ng B10 plasmid, 75 ng B11 plasmid, and 75 ng pCDNA3-GFP1-9 T2A mCherry per transfection) using Lipofectamine 2000 (Invitrogen) according to the manufacturer’s instructions. Four transfections were performed for each combination. Cells were incubated for 48 h, fixed with 4% paraformaldehyde, stained with Hoechst 33342 dye and stored in the dark at 4 °C until imaging. Fixed cells were imaged on a PerkinElmer Opera Phenix High Content Screening System using a 10x/air NA 0.3 confocal lens, acquiring nine fields per well. The excitation and emission wavelengths (Exc/Em) for each channel were: Hoechst 33342 (375/435-480), EGFP (488/500-550), and mCherry (561/570-630). The EGFP channel was scanned independently whereas mCherry and Hoechst were acquired simultaneously. Total cell nuclei, mCherry expressing and GFP expressing cells were detected using CellProfiler software. GFP reconstitution was calculated as the number of GFP expressing cells per those expressing mCherry (measure of transfection efficiency).

To check for construct expression efficiency, cells were transfected with each tagged expression construct together with plasmid encoding GFP B1-10. The GFP1-10 protein spontaneously associates with the B11 peptide. The proportion of cells expressing GFP was then used as a measure of trans-complementation efficiency.

### 2.14. Real-Time Quantitative PCR (RT-qPCR)

The quantification of viral and cellular RNA from both supernatants and cell lysates was performed as described below. In brief, the cell supernatant was collected and centrifuged to eliminate cell debris. TRIzol LS reagent (10296028; ThermoFisher Scientific) was added to the supernatant at a 3:1 ratio. Cell lysate was collected by lysing cells with TRIzol reagent (15596018; ThermoFisher Scientific) after supernatant had been removed and cells had been washed in PBS. Samples were stored at −80 °C until use. Viral and cellular RNA was extracted using phenol-chloroform separation method following the manufacturer’s protocol. RT-qPCR was performed using Luna^®^ Universal Probe One-Step RT-qPCR kit (E3006L; NEB) and amplification was detected and validated by CFX96 Touch Real-Time PCR Detection System (Bio-Rad, Hercules, CA, USA). The cycling protocol was: 55 °C for 10 min, 95 °C for 1 min, followed by 40 cycles of 95 °C for 10 s and 60 °C for 30 s. See Appendix A for primer and probe sequence information [34,35,36,37]. All the primers and probes were synthesized by Integrated DNA Technologies (Coralville, IA, USA). ΔΔCq was calculated relative to control samples following a calculation of ΔCq from each sample based on the Cq of GAPDH. To generate RNA for a standard curve, 612 bp of NP sequence including the RT-qPCR target region described in Appendix A was amplified from a pC-NP plasmid using a forward primer containing a T7 promoter (5′-TAATACGACTCACTATAGGGTCTGTCCGTTCAACAGGG-3′) and reverse primer (5′-ATCACAGCATCGTTGGCATCATG-3′)). After electrophoresis and gel extraction by Monarch DNA Gel Extraction Kit (T1020S; NEB), the PCR product was transcribed using HiScribe T7 High Yield RNA Synthesis Kit (E2040S; NEB) according to the manufacturer’s instructions, followed by degradation of the DNA template by DNaseI treatment (M0303S; NEB) and purification using Monarch Cleanup Kit (T2040S; NEB). The approximate RNA copy number was calculated based on its molecular weight and absorbance measured by a NanoDrop 1000 (ThermoFisher Scientific). For each RT-qPCR reaction set, a five-point 10-fold diluted standard was included to ensure performance of the assay.

### 2.15. In Vitro Pull Down

Maltose binding protein (MBP)-tagged VP40 and His_6_-tagged CAPG were recombinantly expressed in E. coli bacteria. MBP-VP40 and His_6_-CAPG proteins were purified using an amylose or Ni column, respectively, followed by size exclusion chromatography. For pulldown assays, MBP-VP40 (1 mg/mL) was immobilized onto amylose beads prior to incubation with His_6_-CAPG (1 mg/mL). Beads were washed at least three times with buffer containing 10 mM Tris pH 7.5 and 100 mM NaCl. Gel samples were loaded onto SDS-PAGE and stained with Coomassie blue.

### 2.16. Statistical Analysis

GraphPad Prism version 9.0.0 for Mac (GraphPad Software, San Diego, CA, USA) was used to carry out one-way ANOVAs with Dunnett’s or Tukey’s multiple comparisons test, and *p*-values from these analyses were used to determine statistical significance. Significance was taken as *p* < 0.05. All CellProfiler pipelines (method files) used for image analysis are available on Zenodo [31].

## 3. Results

### 3.1. Suppression of CAPG Expression Impairs EBOV Infection

The importance of CAPG for EBOV infection was initially tested by suppressing its expression using four siRNA targeting different portions of the CAPG mRNA. Each reduced CAPG expression ranging from 60 to 80% loss (Figure 1A). Virus yield was measured by challenging cells at an MOI of 0.01, sampling medium after 48 h and measuring virus titer on Vero cells. Compared to non-targeting siRNA, virus yield was greatly reduced (>90%) for each CAPG specific siRNA (Figure 1B). The impact on infection efficiency was confirmed using a recombinant EBOV encoding GFP as an infection marker. Cells were fixed after 48 h, which corresponds to approximately 2 rounds of virus replication after accounting for expression and maturation of GFP [38]. For each siRNA a significant reduction (*p* < 0.001) of EBOV infection, measured by virus protein expression, ranging from 60 to 80% was seen (Figure 1C,D) but was less than seen for the virus yield experiment. Cell viability was confirmed at several time points after transfection of siRNA, and no significant difference was observed compared to non-targeting siRNA (Appendix A). Taken together, our initial tests indicated a role for CAPG in controlling steps early in infection as well as egress but having a greater impact on virus yield from infected cells.

To validate the siRNA findings, CRISPR/Cas9 was used to establish two knockout (KO) cell clones (labeled as #1 and #6) and two partial CAPG-expressing clones (#13 and #14 with >90% loss of CAPG by immunoblot) that mimicked the siRNA induced suppression of CAPG. Each clone had alterations in exon 4, which encodes amino acid residues 66 to 117 and the first actin binding domain (see NCBI NM_001747.4 and ref. [39]). Sequencing indicated clones #1 and #6 had disruption of all alleles while clones #13 and #14 maintained at least one functional allele (Appendix A) and was reflected in loss or reduced protein levels by immunoblot (Figure 1E). Controls were parental Hela cells (WT Hela) and a control line that was CRISPR/Cas9 treated but had no change in CAPG expression (clone control, Figure 1E). Virus spread through the cell monolayer was assessed at different times (up to 72 h) by staining for expression of the EBOV glycoprotein (GP). The two clonal cell lines with low but detectable expression of CAPG showed a 4-fold block in virus infection, with replication remaining low at all time points (Figure 1F). In contrast, the clones with undetectable CAPG expression generally showed normal infection levels with a significant difference observed only at the 24 h time point for clone #6. This suggests these clones had compensated for loss of CAPG expression, which is known to occur during KO selection process for other genes [40]. The growth of each clone was similar to wild type cells (Appendix A). Furthermore, phalloidin staining patterns, measuring F-actin content and morphology in cells appeared similar to wild type cells (Appendix A) suggesting that loss of CAPG did not have global effects on F-actin. Overall, our data indicates that suppression of CAPG by either CRISPR or siRNA results in loss of infectivity, with an apparent greater effect on virus spread in infected cell cultures.

To examine early defects in the virus infection cycle, virion binding and uptake into cells was measured. Virus uptake into cells was measured 6 h after initiation by staining fixed cells with a sensitive GP-specific antibody followed by different fluorescently labeled secondary antibodies before (red) and after (green) permeabilization of the plasma membrane with a non-ionic detergent. The different colored secondary antibodies allowed differentiation of surface bound (both red and green-fluorescent virions) versus those that had entered cells and were inaccessible to labeling until membrane permeabilization (green only). Despite a reduction in internalized virions for siRNA 1 and 2, this change was not significant (Figure 2A). As a second measure of virus uptake, siRNA treated cells were challenged with WT-EBOV and RNA was extracted at 4 h post-infection, a timepoint soon after entry into the cell cytoplasm and when virus mRNA synthesis is first detectable [38]. EIPA and chlorpromazine (CPZ) were used as active and inactive control inhibitors for macropinocytosis (major entry path used by EBOV) and clathrin-dependent endocytosis (not substantially used by EBOV), respectively [9], being treated 1 h before incubation with virus. RT-qPCR was performed targeting the NP gene. Treatment by each siRNA gave no significant change in cell associated virus RNA. In contrast, EIPA, a compound that blocks EBOV uptake by macropinocytosis showed a significant reduction (*p* < 0.01). Chlorpromazine, that blocks clathrin mediated endocytosis, which is not used by EBOV, served as a negative control (Figure 2B). These findings are consistent with CAPG having only a minor role in virus uptake into cells.

### 3.2. CAPG Is Required for Efficient Production of Virus from Infected Cells

Since actin function is known to be important for EBOV budding from cells [20,21] and CAPG regulates actin polymerization, we next tested whether CAPG depletion impacted budding of virions from cells. Viral RNA levels present in cell culture medium was used as a measure of virion release from cells. Signals were compared between viral RNA in supernatants and cell lysates (Figure 3A). While this measurement was complicated by the previously measured reduction in initial infection efficiency, in each case, levels of virus RNA in supernatants were reduced (increase in ΔCq) relative to cell lysate levels and was more pronounced for the weaker siRNA #1 (8-fold reduction) followed by #3 and #4 (each >3-fold reduction). This indicated a defect in virion release that was greater than the reduction in initial infectivity.

Since Ebola virus-like particles (VLPs) can assemble and bud from cells through expression of EBOV VP40 alone, Hela cells were transfected with plasmid encoding VP40 and siRNA targeting CAPG mRNA. VLPs produced from the cells were recovered from culture supernatants by pelleting through a sucrose cushion. VP40 levels in both cell lysate and collected pellets were measured by immunoblot Figure 3B. Compared to VP40 expression in cell lysates, VP40 in supernatants was significantly reduced (*p* < 0.01) by 65-70% in cells treated with each of the strongest acting siRNA (#2–4). This outcome was consistent with the RNA release data indicating that VP40 release from cells and therefore budding of virus from cells is abrogated upon CAPG depletion. In summary, budding of EBOV becomes inefficient in CAPG deficient cells and is reflected in VP40 release as VLPs.

### 3.3. VP40 and GP Are Found in Close Proximity to CAPG in the Cytoplasm

Since loss of CAPG was sufficient to reduce release of VP40 from cells, we tested if CAPG and VP40 associate in cells. To analyze the physical interaction between viral proteins and CAPG, we performed proximity ligation assays (PLA) which are reported to give signal when two targeted proteins are within 40 nm of each other [41] using wild type EBOV-infected Hela cells. As a positive control, we checked amplified signals from NP and VP30 which are known to interact [42]. As shown in the first column of Figure 4A, amplified signals (red) were clearly observed when probed with specific antibodies for each native protein. Similarly, for VP40 and CAPG, strong signals were seen within the cell cytoplasm (Figure 4A, second column). To gage where in the cell this interaction took place, we performed PLA between CAPG and GP. GP is a virus protein that meets with maturing virus particles late in virus assembly. Again, a distinct signal was observed that extended to the periphery of cells (Figure 4A, third column). However, PLA done with CAPG and VP35, a virus protein that becomes part of the nucleocapsid at early steps of virus assembly, gave little signal (Figure 4A, fourth column). These results indicate that GP and VP40 are present at sites that are in close proximity to CAPG. To determine if CAPG and VP40 directly associate, a pull-down assay was performed using purified recombinant VP40 and CAPG. However, no direct binding was detected (Appendix A). One explanation is that CAPG and VP40 co-associate indirectly through actin or the interaction is weak. Indeed, samples stained with phalloidin to detect F-actin showed localization of the CAPG-VP40 puncta along actin filaments and in cell projections that resembled filopodia-like extensions (Figure 4B) and known sites of EBOV budding [13]. 

These results indicate that CAPG localizes with VP40 along actin filaments and is also associated with GP, suggesting a relationship to late stage maturing virions.

### 3.4. Identification of CAPG Subdomains Involved in Actin Interaction Are Important for Association with VP40

CAPG is unique among the Gelsolin/villin family proteins in that it consists of three main structural repeats of single gelsolin-like domains, S1, S2, and S3 (Figure 5A) [43]. To identify subdomains in CAPG important for association with VP40, we used a split-GFP system (Figure 5B) where association of proteins fused to the 10th (GFP10) and 11th (GFP11) β-strands of GFP complements a truncated GFP (GFP1-9) to yield a fluorescent complex [44]. To first establish the system, VP40 was tagged with GFP10 and the cellular proteins CAPG, TSG101 and PABPC1 were tagged with GFP11. TSG101 binds directly to VP40 [45], whereas the polyA binding protein PABPC1 has not been reported to associate with VP40, each serving as controls. GFP10 and GFP11 plasmids were transfected into cells along with plasmid encoding GFP1-9. The number of green (containing reconstituted GFP) fluorescent cells was normalized to the number of transfected cells measured by mCherry expression from the same plasmid.

When full-length GFP11 N tagged-CAPG (N-CAPG) was co-expressed with GFP10-VP40, approximately 15% of cells showed strong green fluorescence (Figure 5C). In contrast, no green fluorescent cells were detected in cells transfected with GFP1-9 alone, GFP1-9 and B10-VP40 (no GFP11), or GFP1-9 plus GFPB10-VP40 and GFP11-PABPC1. The combination of GFP10-VP40 and GFP11-TSG101 consistently yielded more green cells than VP40 plus PABPC1 but was below the threshold for statistical significance in this experiment. Since CAPG produced many more fluorescent cells than TSG101 when co-expressed with VP40, we tested whether other actin binding proteins could associate with VP40 to complement GFP1-9. The CH domain from utrophin (UTRN-CH), was chosen as it binds actin monomers within F-actin and tolerates tagging with GFP [46]. The UTRN-CH domain was fused to GFP B11 and co-expressed with GFP10-VP40. Similar to CAPG, this combination yielded significantly more green fluorescent cells than the negative controls. However, the number of fluorescent cells was approximately half that of CAPG plus VP40.

Having established the tripartite split-GFP system, we investigated which regions of CAPG were necessary for its association with VP40. CAPG domains S1, S2 and S1+S2 were tagged with GFP11 at their amino termini whereas S3 and S2+S3 were tagged at the C-terminus and compared to full length CAPG with similar tags. GFP10-VP40 plus the S1 construct complemented GFP1-9 nearly as efficiently as full length CAPG while S1+S2 nearly doubled the efficiency of GFP expression (Figure 5D). In contrast, the S2 domain yielded fewer fluorescent cells, though still significantly more than the negative controls (Figure 5D). Neither the S3 nor the S2+S3 construct was able to complement GFP1-9 (Figure 5E). As a control for expression and trans-complementation efficiency, the ability of each construct to complement GFP1-10 was measured. S1, S2 and S2+S3 constructs gave similar efficiencies, while S3 and S2+S3 was approximately half that seen for full length CAPG (Appendix A). Overall, these results indicate that the S1 region, which contains one of two actin binding motifs of CAPG [47], is necessary and sufficient for association with VP40 but that association is enhanced by the S2 domain.

## 4. Discussion

The Ebola virus replication cycle is heavily dependent upon actin [9]. Early work with inhibitors of actin polymerization such as cytochalasin D or latrunculin, showed dependence for cell entry by macropinocytosis as well as for virus assembly [9,21]. Similarly, actin-regulating host proteins such as RhoA and the Arp2/3 actin nucleating complex have been identified as necessary for these steps [11,12]. In addition, actin filaments play an important role in transporting the replication complex and interactions with VP40 in the maturing virion [13,20]. Unfortunately, the heavy involvement of actin in many cellular processes makes it difficult to interpret findings produced from direct manipulation of actin. However, recent use of actin-accessory proteins, such as those that bind and stabilize F-actin, has allowed advances in understanding actin function and dynamics [48]. Here, we studied the importance of the actin-accessory protein, CAPG, to gain further insight into the role of actin in EBOV infection. Unlike other accessory proteins, like utrophin, that bind monomers along the actin filament, CAPG is present only on the barbed or growing end of F-actin at the end of the main filament or filament branches [49] and so may offer new insights into how this aspect of actin polymerization impacts virus replication.

The knockdown and knockout experiments showed the importance of CAPG for both virus entry and egress. Although we and others have shown EBOV uptake depends on actin function through macropinocytosis [8,9], we saw only weak impact on this step with particle uptake into cells not significantly affected. Instead, we saw a greater impact on virus egress, consistent with previous reports of actin involvement in this step [21]. Production of both infectious viral progeny and VP40-based VLPs were significantly reduced in the absence of CAPG, suggesting the affected step in replication is late in virus maturation and mediated by VP40. Consistent with this model, CAPG colocalized with VP40 in infected cells and through the proximity ligation assay, CAPG and VP40 formed distinct puncta in infected cells aligned along actin filaments close to the cell periphery. CAPG also associated with the EBOV GP, a protein that associates with VP40 when cell membranes and capsids come together during budding. Similarly, we observed strong signals for VP40 and CAPG in a split-GFP trans-complementation assay, which independently confirmed close association of VP40 with CAPG and demonstrated this was not dependent on expression of other viral proteins or viral replication. This association was mediated primarily by the S1 domain, which can bind and cap actin independently of the S2 and S3 domains. However, direct binding of CAPG to VP40 in pull down assays was not detected suggesting indirect association through another cellular protein.

Actin has been previously found within EBOV virions and VLPs. Although VP40 expression was sufficient to produce VLPs that contained actin, co-expressing GP increased the amount of actin present [21]. A detailed cryo-electron tomography (cryo-ET) analysis of VLPs revealed actin filaments surrounded by a layer of VP40, possibly due to a direct interaction [50]. Our data is consistent with an association of VP40 with actin, but cannot distinguish between direct binding of VP40 to actin or an indirect link via an actin-binding protein. CAPG has been reported to bind tightly to the barbed end and cap actin filaments [47], which would suggest that CAPG cannot be responsible for the distribution of VP40 along the actin filament. However, branching of actin as seen at sites of rapid expansion of membranes produces multiple barbed ends that can be capped by CAPG [49]. Our PLA data showing VP40-CAPG labeling as distinct puncta along phalloidin-stained filaments is consistent with this possibility. Alternatively, VP40 could bind actin directly or use other actin accessory proteins along the filament, which would place it in close proximity to actin binding proteins such as CAPG. We observed a strong signal from VP40 and the utrophin actin binding domain in the split-GFP assay. Utrophin is found on F-actin in a 1:1 ratio with each actin monomer [46]. While we do not know how well the recombinant utrophin binds to actin, other reports show that it can be used to label the entire length of the actin filament [46]. We also note that the signal from VP40 in the split-GFP assay was strongest for the actin binding proteins over the known VP40-interacting protein TSG101, which suggests a more extensive or stable interaction. To more precisely determine the exact site of these interactions, further study will be needed using techniques such as high resolution microscopy as has been recently reported [50].

While additional work is needed to fully understand the CAPG-VP40 interaction and its impact on virus egress, we expect our findings will aid in treatment of EBOV disease. CAPG consists of about 1% of total protein in macrophages, cells that are primary targets of infection and effectors of EBOV disease [51,52]. Therefore, suppressing CAPG function could have significant effects on viral dissemination in patients. Recently, a ROCK inhibitor was shown to indirectly decrease CAPG expression in fibroblast cells and resulted in improved wound healing in heart disease models [53]. It will be interesting to determine if a similar approach may provide improved outcome in a virus disease model.

## Figures and Tables

**Figure 1 viruses-14-01903-f001:**
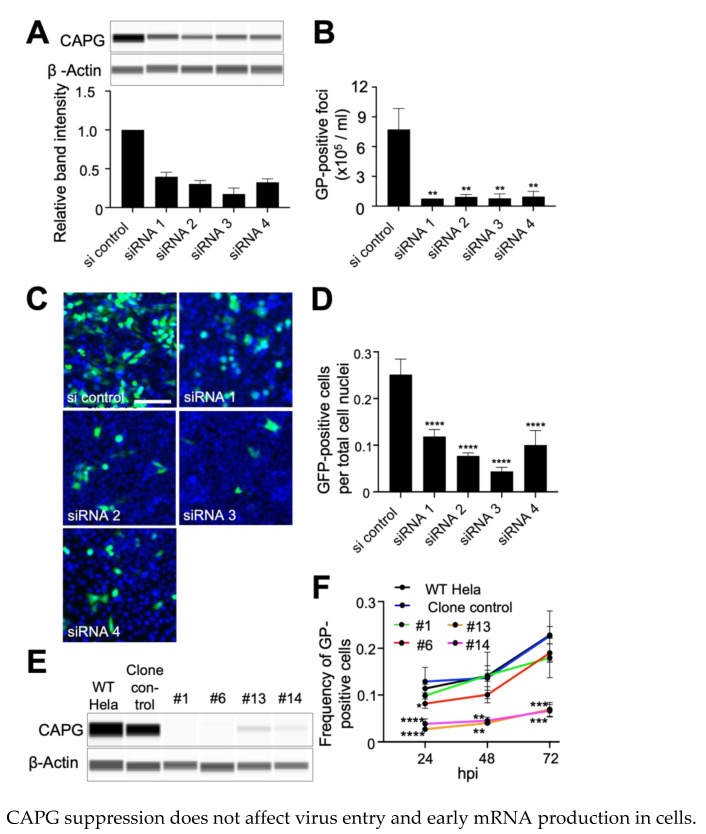
Suppression of CAPG impairs EBOV infection. (**A**) CAPG knockdown (KD) was confirmed by immunoblot assay. Four siRNA targeting different regions of human CAPG were transfected into Hela cells at 40 nM. After 48 h, cells were challenged with virus and a replicate set were lysed for protein expression analysis by immunoblot (a representative blot is shown). β-actin was used as a loading control. Band intensity from triplicate samples was calculated using ImageJ, and normalized to the β-actin loading control. (**B**) Yield of infectious particles in the supernatant was measured by counting GP positive foci on VeroE6 cells. Cells were inoculated with the supernatant from wild-type EBOV-infected cells by limiting dilution. At 48 h after inoculation, cells were fixed and stained with anti-GP antibody and foci counted. (**C**) siRNA treated cells were challenged with GFP-EBOV with representative images shown from 48 hpi. Infected cells expressing GFP (green) and nuclei stained with Hoechst 33342 (blue) are visible. Scale bar = 500 µm. (**D**) Count of GFP-positive cells. The numbers of GFP-positive cells and the nuclei in each image were counted by CellProfiler software and infection efficiency was calculated by dividing the number of GFP-positive cells by that of the nuclei. (**E**) CAPG expression in each knockout (KO) and KD clones was detected by immunoblot. β-actin was used as a loading control. WT Hela = parental WT Hela cells and clones are indicated. (**F**) Time-course analysis of virus spread for each clone. At each indicated time point after infection with WT-EBOV, the cells were fixed and stained with anti-GP antibody. The number of GP-positive cells were counted by CellProfiler software, then normalized to that of the nuclei count in the same image. All data are means of three independent experiments +/− SDs. One-way ANOVA with Dunnett’s multiple comparisons test was used for statistical analysis relative to non-targeting siRNA treated control samples or the CRISPR clone control. *, *p* < 0.05; **, *p* < 0.01; ***, *p* < 0.001; ****, *p* < 0.0001.

**Figure 2 viruses-14-01903-f002:**
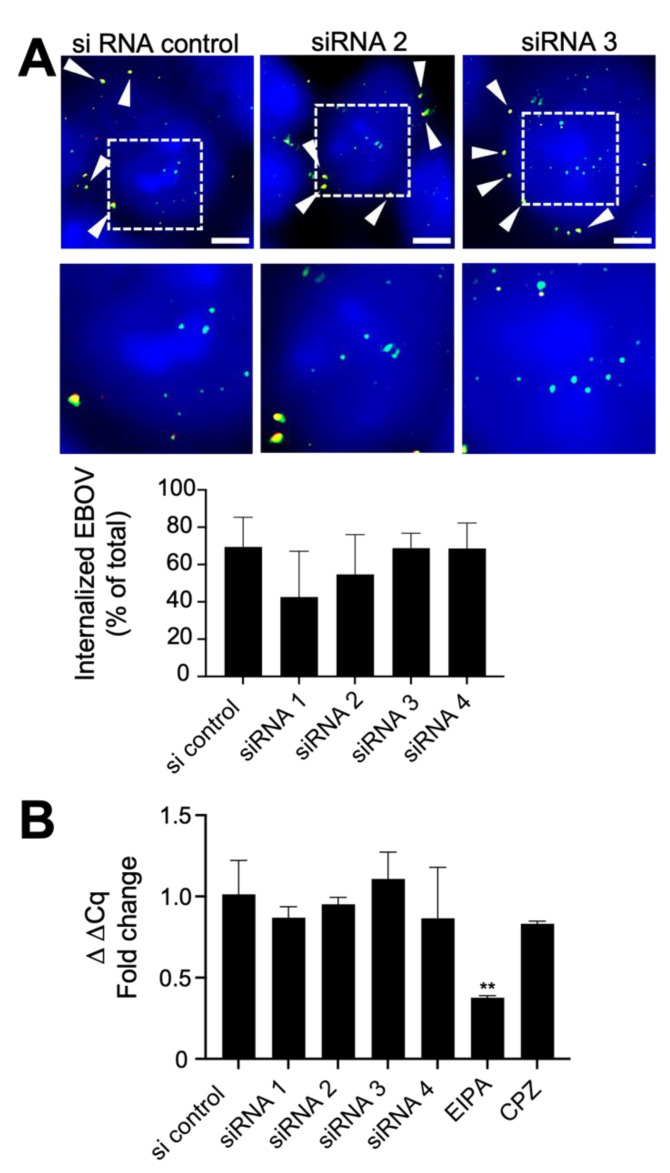
Effect of CAPG suppression on EBOV uptake into cells. (**A**) Hela cells were treated with siRNA for 48 h, incubated with WT-EBOV at 14 °C for 1 h to allow binding without uptake and then temperature raised 37 °C to obtain synchronized virus uptake. Cells were then fixed and stained with anti-GP and Alexa Flour 594 antibody (red). Subsequently, cells were permeabilized with 0.2% Triton X-100 detergent and restained with anti-GP and Alexa Fluor 488 antibody (green). CellMask Blue was used for staining both the cell cytoplasm and nucleus (blue). Arrowheads show double stained puncta (uninternalized virus on the cell surface). Scale bars = 5 µm. Images below are magnified images indicated by dotted lines in areas where fluorescent green stained (internalized) virus particles are evident. The percentage of internalized virus was calculated as the ratio of green foci to total foci and shown in the graph below the images. (**B**) RT-qPCR detection of viral RNA during early infection. siRNA-treated Hela cells were incubated with WT-EBOV for 1 h, then the cells were washed and fresh medium was added onto the cells. The cells were lysed at 4 hpi for RNA extraction. Viral RNA was detected using a primer and probe set targeting NP gene (see Appendix A. ΔΔCq was calculated using GAPDH as a reference control in each sample. The data are shown as fold change relative to siRNA non-targeting control. A 1 h pre-treatment with 50 µM of the amiloride, EIPA, or chlorpromazine (CPZ) were used as macropinocytosis and clathrin-dependent endocytosis inhibitors, respectively. One-way ANOVA with Dunnett’s multiple comparisons test was used for statistical analysis relative to control samples. The means of at least 2 independent experiments are shown ± SD. **, *p* < 0.01.

**Figure 3 viruses-14-01903-f003:**
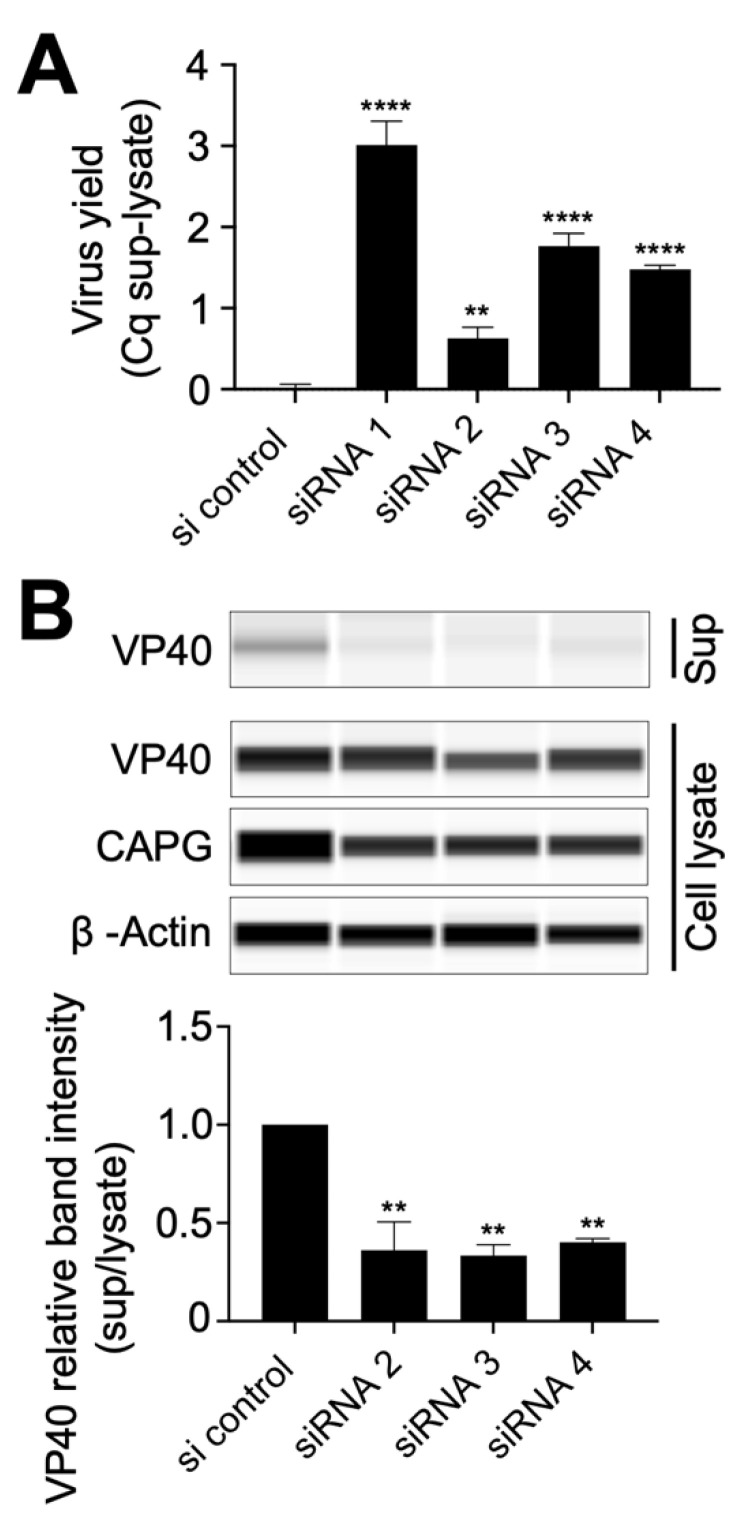
Effect of CAPG suppression on EBOV release from cells. (**A**) Measurement of the quantity of viral RNA released from siRNA treated cells. At 48 hpi, RNA was extracted from the supernatant (sup) and the remaining cells (cell lysate), then virus RNA levels measured by RT-qPCR using primers for NP. The graph indicates Cq supernatant-cell lysate signals in each sample relative to the siRNA control. (**B**) The efficiency of VLP formation from cells treated with each indicated siRNA. Hela cells seeded in a 6 well plate were transduced with siRNA (40 nM each) and pCAGGS-Ebola VP40 plasmid (0.5 µg). At 48 h post transfection, the supernatant was collected and centrifuged to remove cell debris. VLPs were collected by pelleting through a 20% sucrose cushion. VLP pellets and cell lysates were analyzed by immunoblot. Band intensity from each sample is shown relative to siRNA non-targeting control. All assays were repeated at least twice and the representative data sets are shown. One-way ANOVA with Dunnett’s multiple comparisons test was used for statistical analysis relative to control samples. One-way ANOVA with Dunnett’s multiple comparisons test was used for statistical analysis relative to control samples. The means at least two independent experiments ± SDs are shown. **, *p* < 0.01; ****, *p* < 0.0001.

**Figure 4 viruses-14-01903-f004:**
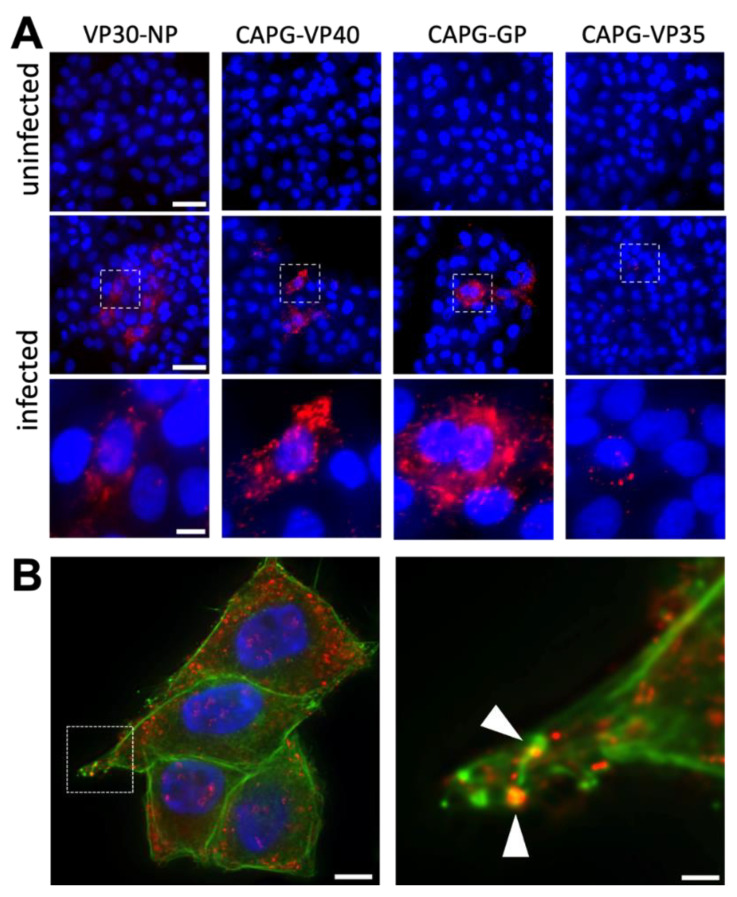
EBOV proteins localize in close proximity to CAPG in the cell cytoplasm. (**A**) Proximity ligation assays were used to detect CAPG and viral protein association. Hela cells were infected with WT-EBOV for 24 h, then fixed and permeabilized. Cells were treated with the indicated antibodies specific for each native protein and detected protein complexes stained red. Cell nuclei were stained with Hoechst 33342 (blue). The lower set of images from infected cells are magnified from the indicated sections of images (dotted squares). Scale bars = 50 µm, and 10 µm in magnified images. (**B**) A representative image (left) of amplified signal (red) of VP40-CAPG and phalloidin (green). The images were taken from the middle z-plane from an image stack. A magnified image (right) is shown for a region indicated by the dotted square. Arrowheads indicate sites where CAPG-VP40 complexes and phalloidin staining (F-actin) overlap. Scale bar = 10 µm and 2 µm in the magnified images.

**Figure 5 viruses-14-01903-f005:**
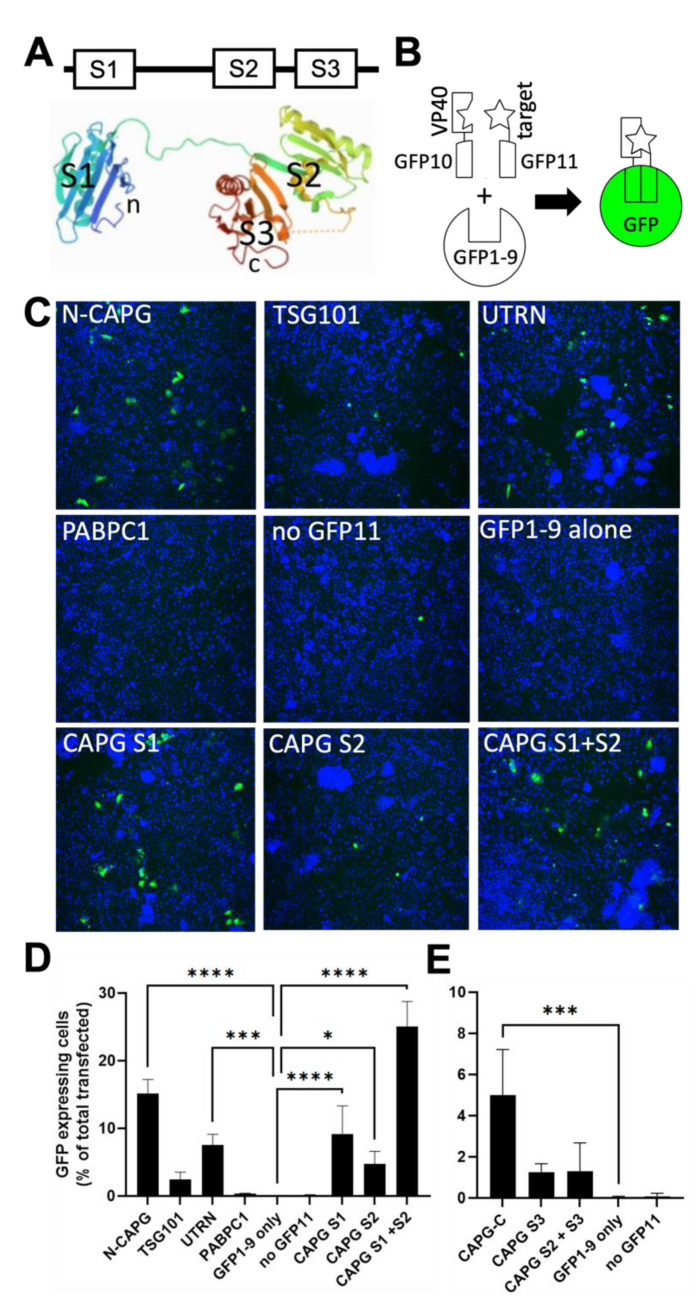
Identification of CAPG subdomains important for interaction with VP40. (**A**) Schematic representation of the CAPG protein. Each gelsolin domain is indicated by S1, S2 and S3. The 3D structure of CAPG is shown and was derived from PDB 1JHW. (**B**) Shows the arrangement of the tripartite split GFP system. For this assay, the GFP10 peptide was used to tag VP40 and the GFP11 peptide was used for the indicated host cell protein targets. When each part is brought together with the remainder, GFP1-9, cells fluoresce green through formation of mature GFP. (**C**) Representative images showing cell nuclei stained with Hoechst 33342 (blue) and GFP positive cells (green), representing reconstituted GFP, for the indicated host protein tagged to GFP11 peptide and VP40 tagged to the GFP10 peptide. (**D**) Quantitation of image sets for N-terminus tagged proteins with GFP11. Sets of 4 wells were transfected with each tagged construct and GFP1-9 expression plasmid that also encoded mCherry as a marker of transfection efficiency. The number of GFP-expressing cells was counted using CellProfiler and expressed as the fraction of transfected cells. (**E**) Quantitation of image sets for C-terminus tagged domains. The S3 domain of CAPG was expressed as a C-terminal fusion with GFP11. Despite lower efficiency of mature GFP formation, significant activity allowed comparison to full length CAPG. Mean ± SD are shown with *, *p* < 0.05; ***, *p* < 0.001; ****, *p* < 0.0001 by One-way ANOVA with Dunnet’s multiple comparisons test.

## Data Availability

Data is available upon request of R.A.D. Pipelines for imaging processing by CellProfiler are available through Zenodo [31].

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
