# Peer review of "CAPG Is Required for Ebola Virus Infection by Controlling Virus Egress from Infected Cells"

_viruses, 2022, doi:10.3390/v14091903_

Round 1

Reviewer 1 Report

The roles of cellular actin accessory proteins during the Ebola virus (EBOV) life cycle are not fully understood. Mori et al. investigate the importance of one such protein, CAPG, during viral entry and egress. They find that RNAi-mediated knockdown of CAPG blunts viral replication, likely owing to reduced viral egress rather than entry. Moreover, they identify an interaction between CAPG and the EBOV matrix protein VP40—essential for viral egress and budding—hinting at a mechanism by which CAPG promotes viral replication. Overall, this is really solid work, and I appreciate the mix of different methods to validate most results in the manuscript. I do wish that they did just one more experiment to really show that their proposed mechanism (the CAPG-VP40 interaction) is correct.

Major comments:

1. In my mind, the main lingering question is: how are assembly and egress affected when CAPG is perturbed? It feels like there’s two separate halves to the manuscript: Fig. 1–3 are entirely CAPG perturbations and phenotypic assays (replication, entry, assembly/egress), while Fig. 4 and 5 show protein-protein interactions (PPI) with native CAPG. It would be really phenomenal to visualize VP40 and viral egress when CAPG is perturbed by RNAi. A likely hypothesis is that CAPG siRNAs cause a trafficking defect such that VP40 is stuck in the cell interior, rather than being transported out to the plasma membrane. PLA for VP40 + GP, or immunofluorescence for VP40, with CAPG siRNA treatment would be a fairly simple way for the authors to demonstrate a mechanism for how CAPG acts as a pro-viral host factor.

Minor comments:

1. Fig. 1E and 1F: the CAPG CRISPR KO cell lines look strange. The authors’ suggestion that the clones compensated for loss of CAPG is possible, but ultimately they never use the cell lines again. Given the possible confusion, and since they are never used in the later figures, I would recommend just dropping the CRISPR KO and KD cell lines entirely.

2. Fig. 4: what type of microscope / what kind of images are shown? The figure legend in 4B implies it is a z-stack from a confocal microscope, but if that’s the case, then there is unexpected PLA signal in the nuclei of cells in 4A. Or is 4A showing a maximum intensity projection? Alternatively, if 4B is not confocal microscopy, then the authors should be careful and avoid saying ‘co-localization’.

3. Fig. 5: one caveat to the split-GFP system is that the data are normalized to an mCherry transfection control on the GFP(1-9) detector plasmid, but not to expression of the individual proteins/domains being tested. Have the authors verified that the individual domains are actually expressed by western blot? If not, please add a sentence in the results as a caveat.

Author Response

Responses follow the reviewer comment and indicated by the "dot"

Major comments:

  1. In my mind, the main lingering question is: how are assembly and egress affected when CAPG is perturbed? It feels like there’s two separate halves to the manuscript: Fig. 1–3 are entirely CAPG perturbations and phenotypic assays (replication, entry, assembly/egress), while Fig. 4 and 5 show protein-protein interactions (PPI) with native CAPG. It would be really phenomenal to visualize VP40 and viral egress when CAPG is perturbed by RNAi. A likely hypothesis is that CAPG siRNAs cause a trafficking defect such that VP40 is stuck in the cell interior, rather than being transported out to the plasma membrane. PLA for VP40 + GP, or immunofluorescence for VP40, with CAPG siRNA treatment would be a fairly simple way for the authors to demonstrate a mechanism for how CAPG acts as a pro-viral host factor.
  • We feel that we tried to do what was asked. If this was not clear, we hope that the changes made, as indicated below, help to clarify the mechanism as far as we were able to determine.

Minor comments:

  1. 1E and 1F: the CAPG CRISPR KO cell lines look strange. The authors’ suggestion that the clones compensated for loss of CAPG is possible, but ultimately they never use the cell lines again. Given the possible confusion, and since they are never used in the later figures, I would recommend just dropping the CRISPR KO and KD cell lines entirely.
  • The cell lines grew similarly to the siRNA treated cells. They also provide an independent verification of the siRNA outcomes. We feel that the CRISPR work is an independent verification of what was seen with the siRNA. In particular, the partial knockdowns appear to affirm the outcome well. So, we would prefer to leave this data in the manuscript.

  1.  what type of microscope / what kind of images are shown? The figure legend in 4B implies it is a z-stack from a confocal microscope, but if that’s the case, then there is unexpected PLA signal in the nuclei of cells in 4A. Or is 4A showing a maximum intensity projection? Alternatively, if 4B is not confocal microscopy, then the authors should be careful and avoid saying ‘co-localization’.
  • We used a widefield microscope but took z-stack images using a 100x lambda lens, which allows optical slicing. The stack is then deconvolved to produce a confocal-like outcome. We have now emphasized this approach in the methods on lines 243-245.

  1.  one caveat to the split-GFP system is that the data are normalized to an mCherry transfection control on the GFP(1-9) detector plasmid, but not to expression of the individual proteins/domains being tested. Have the authors verified that the individual domains are actually expressed by western blot? If not, please add a sentence in the results as a caveat.
  • This is an important point. Protein expression and the ability of each construct to trans-complement GFP reconstitution was measured by co-expressing GFP1-10 along with the GFP11-tagged CAPG or other protein target. The GFP1-10 spontaneously co-associates with the GFP11 tag and fluorescence can be measured per cell. We have now included this data as Figure S5 and made comments on lines 275-278 in the methods and in the results on lines 463-466.

Reviewer 2 Report

Mori et. al. report a new host factor macrophage capping protein (CAPG) required for Ebola virus (EBOV) infection. Authors performed knock down and knock out experiment to show the importance of CAPG for EBOV infection. They show using fluorescence microscopy images, Flow cytometry, qRT-PCR, and western blot of viral protein the entry of the virus is not significantly negatively affected by the loss of CAPG however, the egress of EBOV is significantly altered. Next, they showed that viral protein VP40/GP are in close proximity to CAPG along actin filaments. To corroborate this finding, authors perform split-GFP assay and showed VP40 is in close proximity to CAPG. Furthermore, they show that S1 and S2 domain of CAPG is important for the interaction. 

It's a good manuscript, kudos to the authors! I have few minor concerns which I hope authors can address before publication. 

Minor Concerns:

1. siRNA concentration of 40 nM might induce innate immune response via PKR- and TLR- dependent pathway. I think authors should include this information before showing the knock out assay or discuss this issue. 

2. CAPG knock out clone #1 and #6 showed compensation for the loss of CAPG. It was very interesting but can authors show RNA level for CAPG in these clones? Sometimes the clones might have aberrant version of the protein produced that may not be recognized by the antibody used (epitope might be hidden).

3. To prove direct contact, authors should perform VP40/GP-CAPG pulldown assay. If they had performed it, could you discuss what you found? I'm assuming since authors state "close proximity", authors were not successful in IP-assay. This information would be good to know if the authors performed the IP-assay. 

4. Fig 3A, can be made more clear. I had to read multiple times to understand the experiment and following analysis (line 371-374). Del Cq threw me off. 

5. In fig 4A, please include the first column label in line 394.

6. I highly recommend authors to re-order figure 5 both in text and in figure. It's very confusing. If I was the author, I would include:

Fig B--> Fig C Row 1/2--> Part of D

Fig A--> Fig C Row 3--> Part of Fig D --> Fig E.

Also, please label VP40 somewhere in the figure to indicate the the protein displayed are associated with VP40. 

Author Response

Responses follow the reviewer comment and indicated by the "dot"

From Reviewer 2.

Mori et. al. report a new host factor macrophage capping protein (CAPG) required for Ebola virus (EBOV) infection. Authors performed knock down and knock out experiment to show the importance of CAPG for EBOV infection. They show using fluorescence microscopy images, Flow cytometry, qRT-PCR, and western blot of viral protein the entry of the virus is not significantly negatively affected by the loss of CAPG however, the egress of EBOV is significantly altered. Next, they showed that viral protein VP40/GP are in close proximity to CAPG along actin filaments. To corroborate this finding, authors perform split-GFP assay and showed VP40 is in close proximity to CAPG. Furthermore, they show that S1 and S2 domain of CAPG is important for the interaction. 

It's a good manuscript, kudos to the authors! I have few minor concerns which I hope authors can address before publication. 

Minor Concerns:

1. siRNA concentration of 40 nM might induce innate immune response via PKR- and TLR- dependent pathway. I think authors should include this information before showing the knock out assay or discuss this issue.

  • This is an important point and is the reason that we always include non-targeting controls. Also, other siRNA used at the same dose level had no effect. So, we think innate responses are unlikely to be acting.

  1. CAPG knock out clone #1 and #6 showed compensation for the loss of CAPG. It was very interesting but can authors show RNA level for CAPG in these clones? Sometimes the clones might have aberrant version of the protein produced that may not be recognized by the antibody used (epitope might be hidden).
  • This is a good point that we had thought about. The supplementary data show clear INDELs that would cause a frame shift and would be unlikely to alter RNA levels. The epitope of the antibody is known to be to the N-terminal of the protein.

3. To prove direct contact, authors should perform VP40/GP-CAPG pulldown assay. If they had performed it, could you discuss what you found? I'm assuming since authors state "close proximity", authors were not successful in IP-assay. This information would be good to know if the authors performed the IP-assay.

  • We have performed this assay and included it in figure S4. It shows that we were not able to detect interaction by IP. However, this could mean that we were not able to preserve the interaction. We have added this to the methods on lines 307-314 and in the results 419-421 and in the discussion 502-503.

4. Fig 3A, can be made more clear. I had to read multiple times to understand the experiment and following analysis (line 371-374). Del Cq threw me off.

  • We have relabeled the Y axis to show “virus yield” (Cq sup-lysate) and adjusted the figure legend. We think this makes it clearer.

5. In fig 4A, please include the first column label in line 394.

  • This has been done on current line 408.
  1. I highly recommend authors to re-order figure 5 both in text and in figure. It's very confusing. If I was the author, I would include:

Fig B--> Fig C Row 1/2--> Part of D

Fig A--> Fig C Row 3--> Part of Fig D --> Fig E.

7. Also, please label VP40 somewhere in the figure to indicate the the protein displayed are associated with VP40. 

  • We are unsure if the reorganization of the figure will help with flow. Instead, we have modified the figure legend to help with explaining the figure. We do agree that the VP40 and target should be labeled on Figure 5B. This has been done.